# Evidence of Neuroplastic Changes after Transcranial Magnetic, Electric, and Deep Brain Stimulation

**DOI:** 10.3390/brainsci12070929

**Published:** 2022-07-15

**Authors:** Julius Kricheldorff, Katharina Göke, Maximilian Kiebs, Florian H. Kasten, Christoph S. Herrmann, Karsten Witt, Rene Hurlemann

**Affiliations:** 1Department of Neurology, School of Medicine and Health Sciences, Carl von Ossietzky University, 26129 Oldenburg, Germany; julius.kricheldorff@uni-oldenburg.de (J.K.); karsten.witt@uni-oldenburg.de (K.W.); 2Division of Medical Psychology, Department of Psychiatry and Psychotherapy, University Hospital Bonn, 53127 Bonn, Germany; katie.goke@mail.utoronto.ca (K.G.); m.kiebs@ukbonn.de (M.K.); 3Institute of Medical Science, University of Toronto, Toronto, ON M5S 3G8, Canada; 4Experimental Psychology Lab, Carl von Ossietzky University, 26129 Oldenburg, Germany; florian.kasten@uni-oldenburg.de (F.H.K.); christoph.herrmann@uni-oldenburg.de (C.S.H.); 5Research Center Neurosensory Sciences, Carl von Ossietzky University, 26129 Oldenburg, Germany; 6Department of Psychiatry and Psychotherapy, Carl von Ossietzky University, 26129 Oldenburg, Germany

**Keywords:** deep brain stimulation (DBS), transcranial electric stimulation (tES), transcranial magnetic stimulation (TMS), neuroplasticity, electroencephalography (EEG)

## Abstract

Electric and magnetic stimulation of the human brain can be used to excite or inhibit neurons. Numerous methods have been designed over the years for this purpose with various advantages and disadvantages that are the topic of this review. Deep brain stimulation (DBS) is the most direct and focal application of electric impulses to brain tissue. Electrodes are placed in the brain in order to modulate neural activity and to correct parameters of pathological oscillation in brain circuits such as their amplitude or frequency. Transcranial magnetic stimulation (TMS) is a non-invasive alternative with the stimulator generating a magnetic field in a coil over the scalp that induces an electric field in the brain which, in turn, interacts with ongoing brain activity. Depending upon stimulation parameters, excitation and inhibition can be achieved. Transcranial electric stimulation (tES) applies electric fields to the scalp that spread along the skull in order to reach the brain, thus, limiting current strength to avoid skin sensations and cranial muscle pain. Therefore, tES can only modulate brain activity and is considered subthreshold, i.e., it does not directly elicit neuronal action potentials. In this review, we collect hints for neuroplastic changes such as modulation of behavior, the electric activity of the brain, or the evolution of clinical signs and symptoms in response to stimulation. Possible mechanisms are discussed, and future paradigms are suggested.

## 1. Introduction

Neuroplasticity of the nervous system covers a large variety of phenomena in order to describe changes in the brain on different levels as a reaction to dynamic physiological or pathological conditions. Neuroplasticity can be the result of neuronal reorganization on a molecular, synaptic, and morphometric neuronal level [1]. It can also refer to changes in neural circuits to adapt to changes in the environment (external stimuli) or changes in brain functioning resulting from diseases of the nervous system itself (internal changes) [2]. Several environmental changes induce neuroplasticity such as learning [3,4], sleep [5], aging [6], external stimuli which are accessible to sensory perception [7], and even changes in lifestyle [8]. In addition, a broad range of therapies, ranging from non-pharmacological behavioral therapies [9,10] to pharmacological therapies, induce electrophysiological and neuroplastic changes in the nervous system [11,12]. There is increasing evidence that brain stimulation techniques are beneficial in treating diseases of the brain. Some of these stimulation techniques have been approved by the Food and Drug Administration (FDA), such as transcranial magnetic stimulation (TMS) to treat depression or deep brain stimulation (DBS) to treat advanced Parkinson’s disease (PD).

In this review, we specifically focus on three modalities of brain stimulation techniques, namely TMS, DBS, and transcranial electric stimulation (tES). Although these methods have been investigated for a long time, to the best of our knowledge, their neuroplastic capacity has never been compared in a narrative review. Given the clinical application of these techniques, we focus on studies in humans and refer only briefly to animal models, wherever needed.

For TMS, neuroplasticity is commonly indexed by the change in cortical excitability before and after application of a course of repetitive TMS, which is measured by the amplitude of peripherally recorded motor evoked potentials (MEPs). In order to demonstrate plasticity in brain areas that do not elicit a behavioral reaction, TMS-evoked potentials (TEPs) and neuroimaging techniques can be used. Hence, the neuroplastic capacity of TMS will be subdivided according to the modalities used to reveal neuroplastic changes. Lasting after-effects of TMS have been described as resembling mechanisms of neuroplasticity and as being biologically similar to processes such as long-term potentiation and depression (LTP/LTD) [13,14].

While TMS is a hybrid method that is applied both in the clinic and in experimental settings, DBS in humans is an exclusively therapeutic application. Thus, for DBS, we will focus on clinical signs of plasticity and review DBS-induced neuroplastic changes in three selected disorders, for which DBS has proven to be an effective treatment. On a clinical level, we assume a neuroplastic process to be driven by an effect of neurostimulation, whenever signs or symptoms of a disease (i) improve over a longer course of ongoing stimulation (e.g., weeks or months after stimulation begin) and are stable in the stimulation parameters such as amperage or stimulation frequency, (ii) show a clinically stable course despite stopping the stimulation, and (iii) clinical side effects of neurostimulation occur after a longer stimulation period and are unrelated to disease progression (malplasticity). These considerations can also be transferred to stimulation-related neurobehavioral effects in healthy volunteers when changes in behavior occur over the course of an ongoing stimulation and clearly outlast the period of neurostimulation.

Finally, tES is mainly used in basic and applied research with preliminary clinical applications to date. Transcranial electric stimulation (tES) is an umbrella term and comprises several different techniques, including transcranial direct current stimulation (tDCS), alternating current stimulation (tACS), and random noise stimulation (tRNS). While these techniques are similar in terms of their setup, their effects on neuronal mechanisms and behavioral outcomes differ, and thus, will be discussed separately. To evaluate neuroplasticity, we consider tES-induced after-effects on physiological measurements such as EEG, MEG, EMG, and its effects on observable behavior. Neuroplasticity can occur at different timescales. Short-term plasticity refers to phenomena in the range of milliseconds to seconds which are probably due to neurotransmitter depletion or changes in neurotransmitter influx that modulate the firing rate of neurons. Long-term plasticity operates in the range of minutes to hours and effects can last as long as days, months, or years. For long-term plasticity to occur, changes in NMDA receptor activity, gene expression, and morphology of the synapse are assumed [15]. While brain stimulation also results in short-term plasticity, we focus on effects due to long-term plasticity for the purpose of this review.

A caveat of our selected definition to infer neuroplasticity is that we cannot directly test and evaluate assumptions about the cellular mechanisms of action in the human brain. We assume the cellular basis of these effects to be synaptic changes, including the involvement of AMPA and NMDA receptors, which, in turn, have secondary effects on neurons, networks, and behavior [16]. Hence, we have included studies that investigated the involvement of neurotransmitters known to be relevant for neuroplasticity and that showcased not only neurophysiological and behavioral signs of neuroplasticity but referred also to morphometric changes as a consequence of neurostimulation, whenever available. The reviewed neurostimulation methods are used in different contexts upon which the kind of evidence for neuroplasticity may depend. For example, probing neuroplasticity evaluated by administration of NMDA antagonist is a perfectly feasible approach in healthy young individuals but may be ethically questionable in a sample of diseased patients with chronic DBS. Due to such limitations, finding a general structure that allows us to identify evidence of neuroplasticity for all methods equally well was deemed impossible. Hence, we decided to review the methods in the context in which they are most commonly applied, as described above.

## 2. Transcranial Magnetic Stimulation (TMS)

### 2.1. Overview of TMS Methods

Transcranial magnetic stimulation (TMS) is a non-invasive technique for stimulation of distinct brain regions [13,14,17]. After placing a magnetic coil over a subject’s head, a brief, high-intensity magnetic field pulse can be generated, which, in turn, induces electric currents of sufficient magnitude to depolarize neurons [14]. A single pulse onto the primary motor cortex (M1) can activate the corticospinal tract, thus, inducing a contraction in the targeted contralateral muscle. Using electromyography (EMG), these contractions may be recorded as motor-evoked potentials (MEPs) [18]. As such, single-pulse TMS can be used to map functional cortical representations of muscles or speech functions, an FDA-approved technique known as cortical mapping [19], or can be used as a diagnostic tool, for example, to measure the central motor conduction time in multiple sclerosis [20]. Stimulation of non-motor cortical regions has no directly observable effect, but cortical reactivity to the stimulation can be recorded using EEG, observed in behavior, or subjectively experienced. For example, when applying single pulses to the visual cortex, subjects may report experiencing phosphenes or scotomas [21]. To overcome this issue, coregistration of TMS and EEG has become a successful method of investigating the neural reactivity of brain regions that do not provide an observable behavioral correlate [22]. With the advent of neuronavigation, it is possible to precisely modulate regions across the entire cortex in an individualized manner. Targets localized by functional and structural neuroimaging are becoming widely used and even real-time targeting of the brain’s fiber tracts through tractography-based TMS neuronavigation is currently being developed with promising results [23,24].

In addition to single pulses, TMS can be applied in trains of repeated TMS pulses (rTMS) at various stimulation frequencies to modulate neural activity. Repeated pulses have a more prolonged effect on the brain that outlasts the effects of the stimulation itself by minutes or even hours [25,26,27]. Importantly, rTMS exerts not only local but also distant effects through connectivity between regions, which can be revealed behaviorally, psychophysiologically, or by combining TMS with neuroimaging [28,29,30,31]. These lasting after-effects of rTMS may underlie its successful applications as therapeutic interventions. When rTMS sessions are applied daily over a period of days or weeks, they can produce significant clinical improvement in a variety of neurological and psychiatric disorders [32]. Regarding psychiatric disorders, the majority of evidence stems from antidepressant effectiveness in treatment-refractory major depressive disorder (MDD) [33,34]. TMS further received FDA approval for the treatment of migraine headache with aura, obsessive-compulsive disorder, smoking cessation, and anxiety comorbid with MDD [19] (for a complete overview of current guidelines on the therapeutic efficacy of rTMS, see [32]).

### 2.2. Neuroplasticity

#### 2.2.1. After-Effects of TMS: Changes in MEPs

Because of the lack of an objective and reliable index of cortical excitability outside the M1, early attempts to evaluate TMS-induced neuroplasticity have been largely restricted to M1. Thus, we begin by reviewing neuroplastic evidence after TMS from M1. It should also be noted here that TMS is not only used to induce LTP-like plasticity but also to indirectly probe LTP-like plasticity in the human motor cortex in health and disease and to test the induction of motor cortical plasticity induced by other interventions, for instance, motor training or tDCS [35,36,37,38].

The nature of rTMS after-effects is complex and influenced by many parameters such as the frequency, intensity, and duration of the stimulation. Chen et al. [39] demonstrated that low-frequency (0.9 Hz) rTMS for 15 min (810 pulses, at a stimulation intensity of 115% of MEP threshold) resulted in a significant depression of MEP amplitude for at least 15 min after the rTMS protocol. By contrast, rTMS at 5 Hz, given in separated short trains, has been shown to facilitate motor cortical excitability for at least 30 min [27,40,41,42]. This led to the general assumption that low-frequency (1 Hz or less) rTMS decreases cortical excitability, whereas high-frequency (5 Hz or greater) rTMS increases cortical excitability [43]. However, this assumption has been challenged by a finding that suggests that the intertrain-interval used in high-frequency rTMS protocols is an additional factor, as a continuous stimulation at 5 Hz was found to induce inhibition instead of facilitation [44]. The duration of after-effects is thought to be dependent on the number of pulses given in a protocol, i.e., a higher number of pulses tends to produce greater and longer-lasting effects [42,45]. Nevertheless, it should be mentioned that succeeding studies did not consistently support these results [46] and the direction of effects can even be reversed with varying pulse numbers [47,48,49]. Stimulation intensity is often set as a certain percentage of an individual’s motor threshold (MT), which is defined as the minimum stimulus strength that produces a small MEP (usually 50–100 μV) in the target muscle, in about 50% of 10–20 consecutive trials [50]. Further, it can be distinguished between the motor threshold at rest (resting motor threshold (RMT)) and the motor threshold during slight activation of the muscle (active motor threshold (AMT)). Cortical excitability generally increases as a function of intensity, i.e., intensities less than MT tend to decrease cortical excitability, whereas intensities greater than MT increase cortical excitability [51,52].

More recently, new rTMS protocols that use “patterned” forms of rTMS have been developed. Theta burst stimulation (TBS) is the most commonly used form and consists of bursts of three pulses at 50 Hz, repeated at intervals of 200 ms [53]. This protocol is based on the naturally occurring theta rhythm (5 Hz) of the hippocampus and has been shown to induce synaptic plasticity in animal experiments [54]. TBS can be delivered as a single train of bursts lasting 20–40 s (continuous TBS (cTBS)) which has a primarily inhibitory effect on cortical excitability, for instance, 40 s of continuous TBS (cTBS) reduces the amplitude of MEPs for nearly 60 min [53]. As opposed to that, the burst train can be split up into twenty 2 s sequences repeated every 10 s (intermittent TBS (iTBS)), which has mainly excitatory effects (Figure 1). A total of 190 s of iTBS increases MEPs for at least 15 min [53]. TBS has gained popularity as it induces longer-lasting effects with shorter application time and lower stimulation intensity than conventional rTMS paradigms [55], and has drastically increased time efficiency in its clinical application [56].

Paired associative stimulation (PAS) is another TMS protocol that combines TMS of the motor cortex with peripheral nerve stimulation (PNS) at the wrist [57]. When repeatedly paired with an interstimulus interval (ISI) of 25 ms, it allows for the synchronous arrival of electromagnetic stimulation and afferent (i.e., peripheral) stimulation in the brain, and facilitates cortical excitability. The application of 90 pairs of stimuli (rate 0.05 Hz) led to PAS-induced LTP-like plasticity, which was seen in a long-term increase (<30 min) of the MEP in the target muscle [57,58,59]. By contrast, a shorter interval between the TMS pulse and the PNS pulse (e.g., ISI of 10 ms) led to PAS-induced LTD-like plasticity and a decrease in cortical excitability [59]. Due to the shorter interval, the afferent pulse from the median nerve stimulus arrives shortly after the TMS pulse. Thus, PAS protocols demonstrate some of the characteristics of spike-timing dependent plasticity. In this concept, the precise temporal interval between presynaptic and postsynaptic spikes modulates LTP-like or LTD-like synaptic plasticity [60]. Instead of pairing a magnetic pulse with a peripheral stimulus, another variant is cortico-cortical PAS, which pairs two connected cortex areas with each other by using two TMS coils [61,62]. Moreover, TMS pulses can be paired with subcortical stimulation using implanted deep electrodes in patients with DBS [63]. One very promising novel clinical approach using PAS is the combination of high-frequency TMS of M1 with high-frequency PNS of the contralateral limb as a means of spinal cord rehabilitation. This approach aims to safely induce an LTP-like effect at corticomotoneuronal synapses of the spinal cord leading to improved corticospinal conduction in patients with incomplete spinal cord injury (SCI) [64].

#### 2.2.2. Combining TMS with EEG

Plasticity-like after-effects induced by rTMS were traditionally revealed in the motor cortex in an indirect manner by measuring the change in the MEP amplitude. The combination of TMS with EEG offers an alternative and more direct demonstration of neuroplasticity induced by TMS in humans [65]. Similar to the principles of MEPs, changes in amplitude and latency of the TMS-evoked potential (TEP) can be obtained from the EEG signal across the entire scalp and used to measure changes in cortical excitability [22]. TEPs have proven to be a sensitive and reliable measure of cortical excitability and are comparable to MEP amplitude recordings [66,67,68]. Motor cortex stimulation-induced TEPs are well characterized and differential effects of rTMS protocols on TEPs are largely consistent with those seen in MEPs. Traditional inhibitory protocols seem to produce a decrease in cortical excitability [69,70,71], whereas traditional excitatory protocols seem to increase cortical excitability [65,72]. Measuring TEPs with the combined use of TMS-EEG has several additional advantages over MEPs. First, it has been suggested that it is a more sensitive tool to assess cortical excitability than MEPs, as they are measurable at intensities that are significantly below the motor threshold [70,73,74]. Second, it is also possible to track the spread of activity from the stimulated site to neighboring areas and distant, functionally connected areas, as responses can be recorded from virtually any electrode on the scalp [21,74]. A TMS pulse on M1 in one hemisphere, for instance, spreads ipsilaterally via association fibers but also to the contralateral hemisphere via transcallosal fibers [22,65,75,76]. Third, while MEP paradigms are largely restricted to M1, the combined use of TMS-EEG allows studying after-effects of rTMS on practically any cortical area that is accessible to TMS. For example, TEPs are well defined when TMS is applied to the dorsolateral prefrontal cortex (DLPFC) and plasticity-like after-effects have been demonstrated within this region [67,68,77,78]. Applying TMS to areas other than the primary motor cortex has provided important insights into the generalizability of effects of intensity and duration of rTMS [79]. In addition to these advantages, a limitation of combining TMS with EEG is the risk of various TMS-evoked artifacts in the EEG signal, including TMS-induced muscle, decay, auditory, and blink startle artifacts [78]. However, extensive efforts have been made to minimize these artifacts in TMS-EEG recordings. For a complete overview and challenges of the TMS-EEG methodology, see [79].

In addition to TEPs, rTMS also produces changes in other EEG metrics, such as changes in TMS-evoked oscillatory brain activity and connectivity measures [80,81,82]. Interestingly, ongoing EEG features are now used to provide feedback to determine, for instance, the exact timing of rTMS pulses. Known as closed-loop stimulation, these approaches aim to enhance the neuroplastic capacity of rTMS by coupling the TMS parameters to real-time EEG biomarkers [83,84].

#### 2.2.3. After-Effects Revealed by PET and MR Imaging

Similarly, other neuroimaging studies have demonstrated that rTMS not only induced changes in the area directly under the coil but also in more distant regions of the brain. For instance, rTMS can exert long-distance modulatory effects on subcortical brain regions, including activation of the striatal reward system (e.g., [85]). The magnitude of dopamine (DA) release in response to single rTMS has been shown to be comparable to the administration of d-amphetamine at a dose of 0.3 mg/kg, a compound known to increase synaptic dopamine signaling [86]. In addition to dopaminergic transmission, both the serotonergic system and the cholinergic system also seem to be involved in promoting the after-effects of rTMS [87].

Positron emission tomography (PET) studies have provided additional evidence for neuroplastic changes after rTMS, as it was shown that rTMS of the left M1 influenced the resting activity of the motor system beyond the duration of the stimulation [88,89]. Beyond that, metabolic changes were also evoked in brain regions interconnected with the stimulation site. These studies also demonstrated an acute reorganization of activity to other areas, as movement-related activity in the premotor cortex of the non-stimulated hemisphere increased after inhibitory rTMS, which was interpreted as a compensatory reaction to the inhibitory effect of 1 Hz rTMS [88]. This reorganization of activity probably resembles that in patients after recovery from stroke [90]. A rapid reorganization in functional brain networks induced by rTMS can also be seen using functional MRI. After 1 Hz rTMS to the left dorsal premotor cortex (PMd), a short-term reorganization was seen in the right PMd [91]. Yet, another fMRI study could show that both supra- and subthreshold rTMS over the left M1/S1 influenced the BOLD signal outside of the stimulated area (i.e., supplementary motor area, contralateral M1/S1), while only supra-threshold rTMS increased BOLD signal in the stimulated area [92]. A recent systematic review that presented 33 rTMS studies with pre- and post-rTMS measures of fMRI resting-state functional connectivity (RSFC) demonstrated reliable changes in RSFC after rTMS [93]. Interestingly, the direction of change was not always consistent with the direction traditionally observed in the stimulated brain area. More specifically, conventionally inhibitory stimulation protocols (e.g., 1 Hz) tended to increase RSFC, while the direction of changes after excitatory stimulation protocols was mixed. Moreover, rTMS-induced changes were not confined to the stimulated functional network, but a majority of changes were found in other brain networks. Hence, rTMS effects tend to spread across networks (ibid.). The importance of understanding these relationships can be of particular value, as there is growing interest in attempting to indirectly target distant brain areas through their connections with more accessible cortical areas. To this end, Wang et al. [31] targeted cortical-hippocampal networks by stimulating a subject-specific parietal region that showed high functional connectivity with the hippocampus. Crucially, they were able to demonstrate that increased functional connectivity in these networks positively correlated with associative memory performance after multi-session stimulation. These alterations likely represent neuroplasticity, as the effect persisted for 24 h after stimulation. Similarly, Mielacher et al. [94] augmented iTBS of the DLPFC for MDD treatment by adding daily sessions of stimulation over individualized parietal targets that were functionally connected to the hippocampus and found increased connectivity between hippocampus and DLPFC that lasted days after stimulation.

In addition to functional changes, TMS-induced neuroplasticity has also been demonstrated through structural changes in the human brain, beneath the site of stimulation, as well as in more distal brain regions [95]. Specifically, after a course of standard rTMS treatment for MDD, patients showed increased gray matter density brain volume in the left anterior cingulate cortex which correlated with the clinical response to treatment [96], a finding also shown by measuring MDD patients’ cortical thicknesses in the same region [97]. In a different study, several brain regions were shown to have increased in volume after treatment but this did not correlate to treatment response (left anterior cingulate cortex, the left insula, the left superior temporal gyrus, and the right angular gyrus) [96] as well as an increase in hippocampus volume [98]. Despite the absent relation to treatment response, a corresponding study pointed to another important aspect when considering plastic changes after prolonged rTMS treatment. Noda et al. [99] reported enhanced theta-gamma coupling at the C3 EEG-electrode site to be significantly correlated with hippocampal volumetric change, suggesting a potential structure-function relationship by the rTMS-induced plasticity. However, it should be cautioned that the physical basis of these morphological imaging methods remains poorly defined and seems to reflect tissue characteristics as well as the abundance and distribution of specific cell types (including neurons, glia, vasculature, but also subcellular components such as dendrites and spines) [100].

#### 2.2.4. Pharmacological Evidence

Lasting after-effects of rTMS seem to implicate synaptic changes and are commonly explained by processes that are similar to LTP/LTD plasticity. Probably, the most direct evidence for this assumption stems from an in vitro model of repetitive magnetic stimulation using mouse organotypic entorhino-hippocampal slice cultures. It was found that 10 Hz stimulation not only led to a long-lasting increase in glutamatergic synaptic strength but also increased GluA1 levels as well as enlarged dendritic spines [101]. Since direct evidence is difficult to obtain in human subjects, pharmacological studies can provide essential information about the underlying mechanism of rTMS-induced after-effects by using drugs that act on receptors involved in neuroplasticity. One such study by Huang et al. [102] showed that the use of selective NMDA receptor antagonists interrupted the suppressive effect of cTBS and the facilitatory effect of iTBS. A similar effect of NMDA receptor antagonists was also found on PAS-induced after-effects [58,59] and 1 Hz rTMS [103]. Conversely, the use of d-cycloserine, a partial NMDA agonist, has been shown to further potentiate motor excitability after 10 Hz rTMS [104]. Moreover, it also modulates the effects of TBS-induced plasticity, although here it seems to reverse after-effects of iTBS from facilitation to inhibition [105,106]. A possible explanation for this might be the simultaneous inhibitory and excitatory effects with differing time course. Additionally, PAS- and TBS-induced plasticity were demonstrated to be modulated by calcium channel antagonists [107,108]. Taken together, it appears that the after-effects of rTMS rely on NMDA receptor-mediated glutamatergic function, suggesting that LTP/LTD mechanisms are involved. Ziemann et al. [109] used a temporary ischaemic block of the hand, which produced a reduction in GABA_A_ inhibition in the contralateral motor cortex, to facilitate the induction of plasticity by a low-frequency rTMS paradigm. Comparable to in vitro studies, this finding provides further evidence that the effects of rTMS are due to an LTP-like mechanism.

Even though these studies show unanimously that TMS-induced potentiation and inhibition rely on NMDA receptors, there is mounting evidence that other processes such as neurotrophic, neuroinflammatory, and neuroendocrine factors, or even the neuro-glia network play a role in the observable after-effects (for an overview see [87]).

One exemplary line of evidence comes from studies investigating the brain-derived neurotrophic factor (BDNF) gene. A single nucleotide polymorphism (SNP) on the BDNF gene—BDNF Val66Met—is associated with hippocampal volume, episodic memory, as well as decreased experience-dependent plasticity in the motor cortex in the normal population [110]. Cheeran et al. [111] showed that Val66Met carriers responded differently to cTBS, iTBS, and PAS protocols as compared with Val66Val individuals, suggesting an influence of BDNF on the induction of rTMS after-effects. This, in turn, supports the notion that rTMS truly affects neuroplasticity.

#### 2.2.5. Behavioral and Therapeutic Evidence

The literature reviewed above clearly demonstrates that rTMS elicits after-effects on the brain that outlast the period of stimulation and that these seem to indicate neuroplasticity. However, the exact nature of the after-effects is further complicated by the fact that they interact with voluntary muscle activity and behavioral learning [25] and depend on the history of synaptic activity in the stimulated region, in a manner that is compatible with a concept that is referred to as “metaplasticity” [112]. Metaplasticity is a higher-order form of synaptic plasticity and refers to neuronal activity that primes the subsequent induction of LTP or LTD [113]. A theoretical model for homeostatic metaplasticity is the Bienenstock–Cooper–Munro theory [114], which postulates that the threshold for inducing LTP and LTD is adjusted in response to previous time-averaged levels of postsynaptic activity. Importantly, rTMS plasticity paradigms seem to be consistent with the rules of metaplasticity, as shown in studies using priming stimulation [112,113,114,115]. More specifically, a prior history of increased activity (i.e., induced by another TMS protocol) enhances the effectiveness of inhibitory rTMS protocols, whereas a prior history of reduced activity enhances the effect of facilitatory rTMS [115,116,117,118]. Additionally, motor learning also seems to interact with rTMS after-effects homeostatically [25,119,120]. Such homeostatic interactions are in agreement with the notion that rTMS induces synaptic plasticity.

Ultimately, after-effects seem to also exert influences on behavior and cognition, including cognitive enhancement both in healthy volunteers [121] and in patients suffering from psychiatric/neurological diseases [122].

Neuroplastic changes after rTMS may also underlie the therapeutic benefits of therapy with TMS. The largest body of evidence of clinical effects can be found for treatment-refractory depression, for which most commonly an excitatory stimulation protocol is applied to the left DLPFC [32,123]. Recent evidence favors the use of iTBS protocols, as they have been shown to be clinically non-inferior to conventionally used high-frequency stimulation while allowing for a much shorter application time [56]. Moreover, high-dose (90,000 pulses administered over 50 sessions in five days (10 sessions/day)) intermittent TBS protocols with functional-connectivity-guided targeting demonstrate rapid-acting antidepressant effects even in patients with highly refractory depression [124].

## 3. Deep Brain Stimulation (DBS)

Deep brain stimulation (DBS) was introduced as a treatment for movement disorders in 1987, when A. Benabid implanted deep brain electrodes in the ventral intermediate nucleus of the thalamus (VIM) to treat severe tremor in essential tremor (ET) or Parkinson’s disease (PD) [125,126]. To date, DBS has proven to be effective and reached FDA approval for several indications, such as advanced PD, ET, medication refractory epilepsy, and has gained FDA humanitarian device exemptions for idiopathic dystonia syndromes (iDS) and obsessive-compulsive disorders. The neuroanatomic target structures include the subthalamic nucleus (STN), VIM, the internal part of the globus pallidum (GPi), the thalamic anterior nuclei, and the crus anterior of the internal capsule. A schematic illustration of STN-DBS is shown in Figure 2.

### 3.1. Overview of DBS Methods

The results of the first series of investigations suggest that effective neurostimulation via deep brain electrodes acts like a lesion effect. In his seminal observation of the first patient treated with VIM DBS, Benabid reported that low-frequency stimulation of the VIM in the range of 30 to 50 Hz did not improve tremor but evoked sensory (paresthesias) and motor symptoms (contractions), whereas a stimulation frequency above 100 Hz dramatically improved tremor [125,126]. The VIM was the first target, because previous therapies such as electric or thermic coagulation of the VIM improved contralateral tremor [127], but were irreversible, tissue destructive and often included severe side effects such as sensory loss, paresthesia, or dysarthria. Systemic application of 1-methyl-4-phenyl-1,2,3,6-tetrahydropyridin (MPTP) and local injection of neurotoxin 6-hydroxydopamine (6-OHDA) create a Parkinson’s disease model in animals [128,129], which is the basis for studying the effects of DBS. Electrophysiological studies from animal models of PD and in PD patients demonstrate STN overactivity particularly in the beta frequency [130,131]. An additional chemical lesion in the STN has been shown to lead to an improvement in Parkinson’s symptoms [132]. Translating these findings into clinical research, high-frequency stimulation of the STN in patients with PD reduced signs and symptoms of PD and mimiced the effect of an STN lesion seen in animal studies. Thus, the mechanism of action of DBS was initially believed to be a local inhibition effect (“inhibition hypothesis”). Neuronal inhibition can be explained by a depolarization block in the vicinity of the stimulation, inactivating voltage-gated currents and activating inhibitory afferents, which might be specifically important for GPi stimulation in dystonia [133,134]. In PD, DBS also modifies the firing pattern within the BG, reducing abnormal firing patterns, such as bursts and oscillations in the beta frequency [135,136]. Decreasing the beta frequency within the BG is associated with clinical improvement in akinesia, rigidity, and albeit comparatively weaker, tremor [137]. DBS also excites afferent axons antidromically influencing the motor cortex probably via the hyperdirect pathway [138,139]. A more detailed analysis of cortical stimulation demonstrated a triphasic response pattern within the BG circuits (early excitation, inhibition, and late excitation) [140], DBS of the GPi, and STN inhibit cortical-evoked responses suggesting that it blocks information flow through the GPi (“disruption hypothesis”) [134]. In summary, the mechanisms of action of DBS are not fully understood, may depend on the composition of neuronal elements in the stimulated nucleus, and may depend on the underlying disease-specific pathophysiological conditions. However, these stimulation-induced changes have a network effect, demonstrated by an antidromic effect to the stimulation target afferent fibers, a filtering effect of patterns of oscillation within the BG circuits and downstream effects of efferent fibers influencing the next relay station throughout connectivity. Hence, neuroplastic changes induced by DBS within the nervous system are likely reflected in and observable as network effects.

### 3.2. Studies Demonstrating Effects of Neuroplasticity

#### 3.2.1. Evidence for Neuroplasticity in Essential Tremor

We define clinical evidence for neuroplasticity as improvement of clinical signs and symptoms over time in a constant neurostimulation setting. Constant neurostimulation, in this context, implies that neither volume of activated tissue nor amplitude or stimulation frequency were changed. Clinical signs of plasticity can also be assumed when side-effects of neurostimulation occur over time in a constant stimulation setting. Movement disorders are a suitable candidate to observe clinical effects of neuroplasticity over time. Symptoms are easily observable and allow for a complete and continuous observation of their evolution under chronic DBS. Other disorders, such as for example, epilepsy, where the target symptom of DBS is a reduction in seizure frequency, are much harder to monitor for clinically observable effects of DBS-induced neuroplasticity. Therefore, the disorders ET, PD, and iDS were chosen as examples of DBS-induced neuroplasticity.

The clinical effect of VIM-DBS in ET is two-fold. Initially, it starts as a lesion effect that often substantially improves upper limb tremor in the first days after DBS surgery. When the tremor reoccurs, DBS is initiated, demonstrating an immediate reduction of about 90% in tremor amplitude [141]. This effect is a direct consequence of the disruption of information flow throughout the volume of activated tissue within the VIM and the dentato-thalamic tract, respectively [142]. Most cases have shown a gradual worsening of tremor amplitude over a time frame of years [143,144,145]. The long-term loss of VIM-DBS efficacy may be the result of disease progression and habituation to neurostimulation. Habituation can be interpreted as a neuroplastic effect that diminishes the stimulation effect post electrode implantation and stimulation initiation; both processes are difficult to disentangle. While, ideally, the difference in tremor severity in a stimulation-ON setting between two time points would reflect disease progression alone, habituation effects are likely to contribute as well. Controlling for the rebound effect of the tremor, seen in a third of patients, Paschen and colleagues disentangled the loss of stimulation efficacy over time in a sample of 20 ET patients; 13% of overall worsening in the stimulation-ON condition was attributable to habituation, whereas 87% of worsening in tremor severity could be explained by disease progression [146]. While it is convenient to explain the decrease in clinical efficacy solely in terms of disease progression and habituation, other uncontrolled factors such as prolonged rebound effects or emotional stress during tremor recordings might also have affected the study results. Nonetheless, even if only to a small degree, habituation effects play a role in the time course of treatment by VIM-DBS for severe essential tremor. Patel and colleagues further reported habituation of VIM-DBS in patients suffering from medical refractory tremor in the course of a demyelinating sensorimotor peripheral neuropathy [147]. This observation shows that habituation of VIM-DBS effects is likely not disease specific. While the mechanisms of habituation are presently unknown, they are of interest to prolong the VIM-DBS effect on tremor suppression.

In conclusion, there is a need to understand habituation effects of DBS to identify risk factors associated with habituation and to characterize neuroanatomic structures within the volume of activated tissue. A better understanding of habituation may help to find better stimulation protocols, sweet spots in the area of the VIM, or to develop pharmacological strategies to stop or to delay malplasticity driving habituation. However, to date, there is insufficient data to answer the question if lesion-associated habituation and stimulation-associated habituation share the same mechanisms. This question is of relevance in the treatment of MR-guided focused ultrasound thalamotomy in the context of ET.

#### 3.2.2. Evidence for Neuroplasticity in Parkinson’s Disease

Comparable to ET, STN-DBS in PD patients leads to an immediate improvement in motor functions [148,149]. Tremor, rigidity, and akinesia improve within seconds to minutes. Several studies have documented positive long-lasting effects of STN-DBS in PD even when the neurostimulation is switched off. After medication and stimulation wash-out phase of three to five days, Benabid et al. [150] reported a slight improvement in motor functions six months after surgery as compared with preoperative scores. This observation has been confirmed by other studies [151,152]. Larger clinical follow-up studies have reported equal, or even a slight, improvement in motor function present one to four years follow-up [153,154,155].

These findings are rather unexpected in a progressive disorder. Two studies compared the clinical motor status before electrode implantation six months and four years after electrode implantation and assessed cerebral blood flow (CBF) SPECT. As compared with preoperative baseline and six months after electrode implantation, CBF SPECT four years after surgery demonstrated increased rCBF in the supplement motor area (SMA) in conditions medication-OFF/stimulation-ON [156] and medication-OFF/stimulation-OFF [157]. Changes in rCBF correlated with clinical improvements.

Evidence that the observed rCBF differences are indicative of STN-DBS related neuroplasticity comes from a postmortem study [158] which identified no, or minimal, tissue damage in the vicinity of the electrode tips. This, in conjunction with the observed significant increase in rCBF in the pre-SMA from six months to four years after surgery, would argue against otherwise alternatively hypothesized progressive lesion effects to restore motor functions in a chronic STN-DBS setting.

It also seems unlikely that factors associated with the duration of medication withdrawal or duration of stimulation holidays explain the lack of clinical progression in the medication-OFF/stimulation-OFF condition. Typically, the motor status in PD patients reaches a plateau three hours after switching OFF STN-DBS [159]. Medication and stimulation withdrawal phase vary among studies, often ranging between 10 and 12 h in a majority of studies, in line with the reported studies. The hypothesis of a STN-DBS related neuroprotective effect on DA could also not be confirmed. [123I]FP-CIT SPECT measuring the level of dopaminergic neurons (DA) in vivo, showed a comparable decrease in binding of radioligand in STN-DBS and non-operated PD groups [160]. Another potentially confounding factor might be the patient’s level of physical activity. Stable motor performance significantly increases after surgery.

Investigations about the effect of improved motor performance on physical activity are currently lacking. This is relevant because physical activity interventions are known to be an effective strategy to improve motor symptoms in PD [161]. However, its long-term effect on rCBF in the pre-SMA has not been sufficiently investigated. In conclusion, the mechanisms of a slight beneficial effect of STN-DBS, as described on motor performance, are not known. One explanation beyond the training hypothesis is that STN-DBS induces neuroplasticity that restores pre-SMA function and, to a smaller amount, motor functions in PD.

Long-term STN-DBS is also associated with attenuation of STN resting-state beta band activity. PD is characterized by elevated resting-state beta band activity of local field potentials (LFPs) in the STN. STN-DBS effectiveness is indexed by a reduction in elevated beta band activity. Trager et al. [162] and Chen et al. [163] highlighted that long-term STN-DBS also attenuated beta band amplitudes at rest (stimulation-OFF). Attenuation was already evident three months post implantation and two months post high-frequency stimulation (HFS) start [163] and may be time limited, as no further adaption in the beta band was detected between six and twelve months by Trager et al. [162]. Moreover, it might initially occur broad brand (low- and high- beta band specific) and after two months attenuation might be limited to the high-beta band activity [163]. Lesioning effects, as an alternative explanation, appears unlikely, as attenuation occurred as compared with one month after implantation [163] and was only present in the stimulated STN, in a subset of bilaterally implanted, but unilaterally stimulated patients [162]. While beta band attenuation was associated with overall motor improvement in both studies, the exact relationship between STN resting-state beta amplitude and overall motor improvement is not clear.

Sensory motor integration has also been shown to improve after long-term STN-DBS in patients with PD. Short latency afferent inhibition (SAI) and long latency afferent inhibition (LAI) index different aspects of sensory motor integration, likely with different anatomical origins (see Turco et al. [164] for an overview); both refer to effects of cortical inhibition of sensory evoked potentials, and are impaired in PD [165]. Sailer et al. [165] found SAI to be impaired in PD only ON dopaminergic (MED-ON) medication as compared with HC, whereas LAI has been shown to be impaired regardless of medication status. Shukla et al. [166] assessed the long-term effects of STN-DBS on SAI and LAI, considering effects of DBS (ON/OFF) and dopaminergic medication (ON/OFF) over time. One month post implantation, the effects of STN-DBS were difficult to discern. However, six months post implantation, DBS-ON was able to offset the impairment on SAI caused by MED-ON. Furthermore, LAI normalized under DBS-ON in conjunction with MED-ON after six months. Proprioception deficits present under MED-ON improved in conjunction with DBS. However, the relationship between LAI and SAI improvements is not clear.

At present, long-term volumetric effects of long-term STN-DBS are lacking. To the best of our knowledge, only two studies have assessed volumetric changes after long-term STN-DBS retrospectively in a sample of PD patients undergoing staged bilateral implantation [167,168]. While both studies reported volumetric changes, they were in disagreement on whether long-term STN-DBS overall led to volume reductions or increases in the targeted brain structures. Sankar et al. [167] and Kern et al. [168] both found reductions in hippocampal volumes, although in different hemispheres (both hippocampi or only ipsilateral to the stimulated STN) and to different degrees (~14\% and ~3\%). The large decrease in hippocampal volume observed by Sankar et al. [167] was not accompanied by a decrease in neuropsychological measures. With regard to BG structure volumetric changes, both studies were in disagreement. Sankar et al. [167] reported increases in putamen volume contralateral to the stimulated STN, Kern et al. [168] found overall decreases in basal ganglia-thalamocortical circuits (includes caudate, putamen, pallidum, and thalamus), particularly ipsilateral to the stimulated STN. The disagreement in the results of both studies may partially be explained by variable stimulation durations, low imaging resolution (1.5T), and small sample sizes. While both studies differ in results and methodology, in conjunction, they illustrate that long-term STN-DBS might also affect brain volume

In terms of long-term neuroplastic functional changes, a longitudinal study by Ge et al. [169] assessed alternations of the PD-related metabolic covariance pattern (PDRP) expression using F-FDG PET in a group of healthy control participants, PD patients, and PD patients (N = 9), who underwent STN-DBS. DBS participants were scanned pre-operative, three, and twelve month post operation, with post-operation scans being performed OFF-Meds and OFF-DBS. The therapeutic decrease in UPDRS scores at three month post operation was associated with a reduction PDRP. Moreover, graph theoretical network analysis of the F-FDG PET images showed that the initally increased small-worldness coeffcient within the PDRP subspace (as compared was healthy controls) was normalized three month post operation. This illustrates the capacity of DBS to exert long-term effects on functional network organization.

Mechanisms by which neuroplastic/neuroprotective effects of STN-DBS in PD occur are presently only well investigated in animal models. Preclinical work suggests a prominent role for BDNF inducing neuroplastic changes, in the nigrostriatal system and the motor cortex (e.g., [170]). However, clinical confirmatory evidence is still not available. On the contrary, BDNF rs6265 polymorphism in Parkinson’s patients has been shown not to be predictive of clinical outcome two years post STN-DBS [171].

#### 3.2.3. Evidence for Neuroplasticity in Dystonia

The GPi is the established stimulation area to treat dystonia. As compared with ET or PD, the evolution of the antidystonic effect after GPi-DBS differs. Whereas phasic movement patterns respond fast after switching on neurostimulation [172], tonic postures and patterns improve only over weeks or even months after stimulation onset. Most studies have demonstrated a plateau in the treatment effect after three months [173]. Therefore, it is plausible that a long-lasting neuroplastic effect occurs, which leads to an improvement in symptoms over time. Dystonic symptoms may recur rapidly when the stimulation is discontinued in the first years [174]. However, in some cases, it has also been observed that after cessation of long-term stimulation, the therapeutic effect remained sustained over time [175,176]. Among the potential explanations for these neurological benefits, it can be assumed that DBS therapy may have the capacity to induce plastic changes that lessen or obviats the need for further treatment [177].

Whereas in PD patients reduced neural plasticity is often observed [178], dystonia patients exhibit excess neural plasticity [179,180,181]. Ruge et al. [180] tested long-term effects of GPi-DBS on neural plasticity via paired associative stimulation (PAS) and short-latency intracortical inhibition (SICI). SICI was increased as compared with HC pre-DBS implantation (indexing reduced inhibition) and reached normal levels over the course of treatment (one-, two-, and three-months post implantation). Response to PAS was also increased as compared with HC prior—indexing increased plasticity. Whereas SICI measures reduced gradually over the course of treatment, responses to PAS dropped sharply below HC response before gradually returning to normal levels as compared with HC. Increased plasticity preoperative has also been shown to correlate with symptom severity and benefit of DBS three months post implantation [181].

Ni et al. [182] linked GPi-DBS and the normalization of neural plasticity directly. In dystonia patients who had received clinically effective DBS (at least six months prior), single pulse GPi-DBS (only at clinically effective contacts and dosages) resulted in two distinct evoked potentials (EP) in the motor cortex. A negative EP at ~10 ms and a positive EP at ~25 ms with specific facilitatory and inhibitory effects on motor evoked potentials (MEP). In combination with TMS, if the interstimulus interval was 10 ms, MEP amplitudes were relatively increased. In contrast, if the interstimulus interval was 25 ms, MEP amplitudes decreased. Single pulse GPi-DBS and PAS with interstimulus intervals at 10 ms or 25 ms lead to motor cortical facilitation during and 30 min post PAS. The effect was more pronounced at interstimulus intervals of 25 ms. While these effects occur presumably after normalization of initially observed hyperplasticity [180], Ni et al. [182] provided evidence for the causal relevance of GPi-DBS for cortical plasticity in dystonia.

## 4. Transcranial Electric Stimulation (tES)

### 4.1. Overview of tES Methods

Transcranial electric stimulation refers to a variety of methods where a small electric current is non-invasively applied to the brain via two or more electrodes on the scalp. The current flows through the skin, bone, and brain tissue from one electrode to another. Depending on the precise stimulation parameters (particularly the waveform), tES can be subdivided into different techniques among which transcranial direct current stimulation (tDCS), transcranial alternating current stimulation (tACS), and transcranial random noise stimulation (tRNS) are the most commonly used to date. As compared with TMS and DBS, tES methods are relatively young and no reliable stimulation protocols have yet been established for successful therapy of neurological or psychiatric conditions, although first trials are underway to move the methods towards clinical applications (for an overview of clinical research on tDCS, see Zhao et al. [183], for an overview of clinical research on tACS, see Elyamany et al. [184]).

Due to skin sensations in response to electric stimulation, the intensity of tES is limited to one or a few mA (milliamperes). This results in an important difference to DBS and TMS, which are considered to be super-threshold brain stimulation techniques, i.e., the electric field resulting from stimulation can directly excite or inhibit neurons to fire action potentials or suppress firing, respectively. In contrast, tES is considered to be a subthreshold brain stimulation technique, since the electric field inside the brain is comparably weak and will only modulate the likeliness of neuronal firing in case of tDCS or the spike timing in case of tACS. The spike rate is not directly manipulated.

It is important to know where to place the stimulation electrodes on the scalp in order to target a specific brain region. For this purpose, finite element models have been used to predict the pattern of current density resulting from electric stimulation. At first, T1- and T2-weighted images are acquired with magnetic resonance imaging (MRI). Then, the images are segmented into tissues of different conductivity. Lastly, a computer algorithm (e.g., the Roast toolbox in MATLAB or SIMNIBS) computes the pattern of current flow for the selected electrode montage (as shown in Figure 3). Recently, we were able to demonstrate that a high correlation of the pattern of current flow and the source localization of the brain activity that was intended to be modulated by tACS resulted in stronger effects of amplitude enhancement (Kasten et al. [185]).

### 4.2. Assumed Neuronal Mechanisms

#### 4.2.1. tDCS

During tDCS, a static electric field is applied between two or more electrodes for a duration commonly ranging between 5 and 20 min. The static field induces a subtle shift in a neuron’s resting membrane potential. Depending on its polarity, neurons exposed to the electric field are slightly de- or hyperpolarized, increasing or decreasing the likelihood of firing an action potential in response to incoming post-synaptic potentials [186,187,188].

In humans, the effects of tDCS were first investigated by assessing the size of motor evoked potentials (MEPs), which occur in response to single-pulse TMS over the motor cortex. Generally, it has been observed that the size of MEPs increased after anodal stimulation of the motor cortex, whilst decreasing after cathodal stimulation [35,189]. This has led to the notion that tDCS increases cortical excitability below the anode, while decreasing excitability below the cathode. However, more recent modeling work has pointed out that the exact effects on the neuronal level are highly dependent on the neuron’s orientation relative to the applied electric field, which is strongly determined by cortical folding [190]. In addition to acute effects during stimulation, it has often been observed that physiological effects persist for several tens of minutes after tDCS is switched off [35,189]. The duration of these effects depends on the duration of tDCS application. While stimulation durations of 5 to 7 min induce only short-lived after-effects in the range of 1–5 min, durations of 9–13 min can induce long-lasting alterations of MEPs for 30 or even up to 90 min [189]. In addition to these initial benefits of increasing tDCS dosage by longer stimulation durations, several studies have report an overall nonlinear relationship between tDCS dose and after-effects when stimulation amplitudes and/or durations are further increased. For example, the excitatory effect of anodal tDCS has been observed to revert after prolonged application for 26 min [191]. Similarly, the inhibitory effect of cathodal tDCS was reversed after 20 min of application when stimulation intensity was increased from 1 mA to 2 mA [192]. A more recent systematic comparison of stimulation amplitudes of anodal and cathodal tDCS, additionally, found no evidence for a correlation of stimulation current and the strength of tDCS after-effects on MEPs [193]. Interestingly, after-effects of tDCS do seem to accumulate in protocols that utilize temporally spaced, repeated sessions of stimulation [194,195], suggesting a possible involvement of late-stage LTP-like plasticity [191].

Although tDCS effects are most widely studied with respect to their influence on MEPs, tDCS has been shown to affect more direct measures of human brain activity such as eliciting lasting changes in EEG activity. It has been suggested that, due to its effect on cortical excitability, tDCS modulates EEG oscillations which sources are located in the stimulated target regions in the brain. However, which frequency band in the EEG is affected by stimulation as well as the direction of these effects seem rather inconsistent, difficult to predict, and may vary depending on the background task. For example, Miller et al. [196] reported a reduction in frontal-midline theta band power following anodal tDCS during a sustained attention task, while Zaehle et al. [197] reported a decrease in theta band power after cathodal tDCS, accompanied by similar spectral changes in the alpha band during a working memory task. When targeting regions in the motor cortex, Ardolino et al. [198] reported increased power in the delta and theta band following cathodal tDCS, while Matsumotu et al. [199] found that anodal tDCS increased event-related desynchronization of motor cortical mu-oscillations, while cathodal tDCS reduced it. Again, the two studies differ in terms of the underlying background tasks (rest vs. motor imagery). When targeting posterior brain regions during rest, Spitoni et al. [200] found a positive effect of anodal tDCS on spectral power which was limited to the alpha band, but no effect of cathodal stimulation. A similar effect was later reported by Mangia et al. [201], who, however, also reported a significant effect of anodal tDCS on power in the beta band. Interestingly, EEG changes elicited by tDCS are commonly only observed for a few minutes after stimulation, which is in contrast to the partly hour long effects observed on MEPs.

Another line of evidence comes from the investigation of changes in brain connectivity as a consequence of tDCS (for a review, see [202]). For example, resting-state data have been recorded with functional magnetic resonance imaging (fMRI) before and after tDCS stimulation. Participants who received stimulation revealed significant changes in regional brain connectivity in the default mode network and in fronto-parietal networks as compared with participants who received sham stimulation [203].

TDCS has also been applied as a therapeutic tool in multiple diseases. For example, tDCS has been demonstrated to suppress the symptoms of depression (for review, see [204]). An evidence-based analysis reported that tDCS was probably also effective to treat symptoms of fibromyalgia and addiction/craving [205].

While acute effects of tDCS during stimulation have been linked to shifts in membrane polarization, offline effects of tDCS are usually explained by processes of LTP- and LTD-like synaptic plasticity (for an overview see Stagg and Nitsche [206]). In particular, it has been shown that selective NMDA receptor antagonists reduce or completely abolish after-effects of anodal and cathodal tDCS on motor cortical excitability in vivo and in vitro [207,208]. Evidence from in vitro stimulation of slice preparations of mice further suggested an involvement of the brain-derived neurotrophic factor (BDNF) [208,209], which was involved in all stages of NMDA receptor-dependent LTP, whereas its precursor peptide (pro-BDNF) has been associated with LTD [210,211]. More recently, it was observed that a frequent polymorphism in the BDNF gene (nonconservative amino acid substitution from valine (Val) to methionine (Met) on codon 66) modulates the size of after-effects of anodal (but not cathodal) tDCS [212]. In that study, participants with the Val66Met polymorphism showed stronger enhancement of MEPs after tDCS as compared with participants with Val66Val after about 20 min post stimulation. Interestingly, this finding is contrary to the effect of the polymorphism on after-effects in other stimulation methods, where participants carrying the Val66Met polymorphism tended to show reduced or even abolished responses to the stimulation protocol (e.g., iTBS [212] and tACS [213]). Delivering an NMDA agonist prior to anodal tDCS of the human motor cortex increased the duration of enhanced MEPs from one hour to one day [214]. Long-lasting after-effects of tDCS on motion perception have even been found to persist over a time period of 28 days [215].

#### 4.2.2. tACS

The effects that tACS has on ongoing brain oscillations during stimulation are believed to rely on neural entrainment, while the after-effects that outlast the end of stimulation are assumed to be implemented by neural plasticity [216]. By definition, entrainment itself does not outlast the stimulation period. Nevertheless, the effect of entrainment does not vanish instantly. For a few cycles after stimulation offset, the internal phase of the oscillation is still coupled to the external force, as has been reported for rTMS [217] and tACS [218]. After-effects indicate that tACS interferes with cortical neurons and demonstrate the efficacy of the method with potential for therapeutic applications.

It has been demonstrated that 10 min of tACS at an individual’s EEG alpha frequency (IAF) resulted in enhanced EEG alpha amplitudes in a time window of three minutes after the end of stimulation [219]. In order to investigate the duration of this after-effect, another study recorded 30 min of EEG after 20 min of tACS at IAF and observed elevated alpha amplitudes for the whole time interval after stimulation had ended [220]. Interestingly, the effect could only be observed when participants had their eyes open and started out with low alpha amplitudes, but not when they had their eyes closed and started out with high alpha amplitudes. Since that after-effect was still observable even at the end of the recording interval, yet another study recorded EEG for an even longer time period after stimulation, i.e., 90 min and demonstrated that 20 min of tACS achieved an after-effect on EEG alpha oscillations for 70 min [221].

Importantly, the after-effect of enhanced EEG alpha amplitudes also modulates cognitive processing after the end of stimulation. For example, improved mental rotation ability has been observed for about one hour after the end of tACS at individual alpha frequency over the visual cortex [222]. Along the same lines, tACS in the alpha frequency range improved multiple other visual processing abilities [223,224]. If other brain regions are stimulated, alpha-tACS can also achieve after-effects on other cognitive functions such as word processing in the prefrontal cortex [225] or motor behavior over the precentral cortex [226].

Notably, after-effects of tACS were not always detected [227], indicating that the phenomenon possibly depends upon stimulation intensity and/or duration. In line with that finding, it has been demonstrated that one second of tACS was not sufficient to achieve any after-effects suggesting a dose-response relationship [228]. This is in line with animal experiments that stimulated for a few seconds and were able to demonstrate entrainment but no after-effects. Another form of dose-response relationship has been demonstrated more recently by studies that were able to relate the strength of tACS-induced neurophysiological and behavioral after-effects to individual differences in the applied electric field, which could vary substantially due to anatomical differences [185,229].

It should be noted that after-effects of tACS on brain oscillations are not only observed on EEG/MEG amplitudes but have also been demonstrated for other parameters such as, for example, phase coherence between hemispheres probably reflecting changes in functional connectivity [230,231].

Note in addition that entrainment and plasticity are not mutually exclusive and may rely on each other [227]. It is plausible to assume that a successful entrainment during stimulation might be a necessary requirement for the generation of neuroplasticity reflecting enduring after-effects. The first evidence for the assumed interaction of online entrainment and after-effect was reported by [232]. These authors were able to demonstrate that the strength of an increased alpha amplitude after the end of stimulation correlated positively with the power during stimulation. These findings suggest a relationship between entrainment and plasticity, in which stronger entrainment predicts stronger after-effects.

However, Vossen et al. [227] showed that entrainment may not be required to produce tACS after-effects. They applied short durations of tACS at individual alpha frequency with short breaks of an equal duration. The experiment was composed of four conditions: short/phase continuous (i.e., no phase shifts between trains of stimulation) with three seconds of stimulation and three seconds of break; long/phase continuous with eight seconds of stimulation and eight seconds of break; long/phase discontinuous with eight seconds of stimulation and break, and phase shifts of 0, 90, 180, or 270 between trains of stimulation, as well as a sham condition with only one train of stimulation at the start of the experiment. The authors compared pre- versus post-stimulation EEG periods and reported a significant increase in alpha power for the long stimulation trains as compared with short stimulation trains and sham. The increased after-effect was observed irrespective of the continuity of phase, suggesting that entrainment was not required for after-effects.

Clinical studies using tACS have revealed that stimulation in the gamma frequency range can improve memory scores in Alzheimer’s disease [233]. In patients suffering from schizophrenia, tACS in the alpha frequency range was successful to decrease auditory hallucinations [234,235]. For reviews on further long-term effects of tACS in clinical populations, see [184,236].

Animal experiments can investigate synaptic plasticity by stimulating the pre-synaptic neuron and recording from the post-synaptic neuron. Such experiments have revealed that synaptic weights change depending upon the relative timing of pre- and post-synaptic spikes, a rule that is referred to as spike-timing dependent plasticity ((STDP) see Figure 4a). A simulation using artificial neural networks has tested whether this synaptic mechanism was susceptible to repetitive stimulation and could be responsible for the after-effects of tACS [219]. As shown in Figure 4b, neurons were interconnected with axons of different axonal delay times representing different resonance properties of the established neuronal loops. When this network was stimulated with a spike train of 10 Hz and synapses were updated according to the STDP rule, the synapses were strengthened that were incorporated in loops with resonance frequencies at the stimulation frequency and slightly above (Figure 4c).

All of the above evidence is, of course, only indirect evidence for synaptic plasticity. It would be desirable to see that neurotransmitters known to be involved in plasticity play a role in the observed after-effects. A recent study demonstrated that when an NMDA receptor antagonist was given to participants, the after-effect of 20 Hz tACS on motor cortex excitability and EEG beta oscillations was abolished [237]. Another finding that pointed in the same direction investigated the effect of a genetic polymorphism of the brain-derived neurotrophic factor (BDNF) [213]. The authors were able to demonstrate that the observed increase in the EEG alpha amplitude after stimulation with alpha-tACS as compared with a control group was a function of the Val66-Met polymorphism of the BDNF gene.

#### 4.2.3. tRNS

tRNS uses band-limited white noise as a signal for electrical stimulation. The effect of tRNS is believed to be due to modulation of ion channels and/or the noise raising the peaks of subthreshold neural oscillations above the threshold for firing, a mechanism referred to as stochastic resonance [238]. It has been demonstrated that 10 min of tRNS of the motor cortex led to enhanced TMS-evoked motor potentials for up to 60 min after the end of stimulation [239]. The effect also seems to be dose dependent, i.e., five and six minutes of tRNS result in after-effects, while 4 min of stimulation are not sufficient [240]. In contrast to the after-effects of tDCS and tACS, the after-effect of tRNS is not modulated by NMDA receptor agonists or antagonists but is suppressed by the GABA agonist lorazepam [241]. Interestingly, the BDNF polymorphism (Val66Val/Val66Met) that has been suggested to modulate the induction of LTP-like plasticity in other brain stimulation methods such as iTBS, anodal tDCS, and tACS, has not been found to influence tRNS after-effects on MEPs [212].

The fact that tRNS achieves after-effects on oscillatory EEG activity [242], despite its inability to entrain brain oscillations due to its non-rhythmic pattern [238], supports the abovementioned notion that entrainment may not be required for synaptic plasticity effects of tES.

## 5. Comparison of Methods Regarding Neuronal Plasticity

In this narrative review, we present three stimulation methods, i.e., TMS, DBS, and tES, and summarize the evidence suggesting a neuroplastic capacity of these neurostimulation techniques. 

For TMS, evidence suggesting that after-effects are produced through neuroplastic mechanisms comes from three types of results: (i) rTMS protocols induce changes in cortical excitability, as seen in MEP amplitudes, which outlast the period of stimulation; (ii) lasting after-effects on brain activity after rTMS protocols can also be revealed using neuroimaging; (iii) the effects of rTMS are altered by drugs that act on receptors involved in neuroplasticity, for esample, NMDA receptor antagonists. As a restriction, it must be stated that patients in clinical brain stimulation studies are often also treated pharmacologically (e.g., pharmacological treatment kept stable), hence, reported long-lasting effects might be favored by metaplastic phenomena induced by chronic pharmacological treatment [243].

For DBS, the time course of the evolution of symptom relief after switching on GPi stimulation for dystonia is compatible with the assumption of a neuroplastic effect. Especially tonic patterns of a dystonic syndrome improved after weeks or months of active stimulation. In some patients, it has been observed that during stimulation cessation, after long-term stimulation, the therapeutic effect is sustained over time. In VIM-DBS for ET, habituation may reflect neuroplasticity, whereas in STN-DBS, hints toward evidence for neuroplasticity come from electrophysiological studies reporting after-effects of stimulation in the beta band range of oscillatory activity.

For transcranial electric stimulation, the evidence for after-effects comes mainly from three types of results: (i) TMS-induced MEPs reveal modulations of cortical excitation during electrical stimulation of the motor cortex, these changes outlast the end of the stimulation period; (ii) parameters of EEG and MEG oscillations such as amplitude and phase coherence have been observed to be elevated after the end of stimulation lasting for up to an hour; (iii) behavioral changes such as reaction times or error rates in cognitive experiments induced by tES that outlast the end of stimulation as well as reduced symptoms in neuropsychiatric diseases. At least the first two types of results are also modulated by neurotransmitters known to be involved in synaptic plasticity.

All three brain stimulation methods reviewed here reveal indirect signs of neuroplasticity, i.e., after-effects of elevated EEG/MEG amplitudes as well as behavioral or clinical after-effects. The three stimulation techniques differ in terms of the volume of tissue activated. We hypothesize that the neuroplastic effects are mediated by different mechanisms, i.e., how these stimulation techniques influence brain networks. DBS stimulates a focal circumscribed volume of tissue, acting as a hub in a neural network. Small brain nuclei such as STN and VIM circumscribe fiber tracts, such as the ansa lenticularis, are powerful interfaces within the motor network. Network effects, therefore, arise from a focal manipulation of network hubs within a deregulated neuronal system; tES and rTMS are less focal neurostimulation techniques. However, their potency to change EEG oscillations argues for their impact to influence brain network functions. Taken together, all three stimulation techniques have a capacity to interfere with brain networks and modify neuronal network functions. 

Importantly, it has also been demonstrated that genetic polymorphisms and drugs that affect the function of neurotransmitters responsible for synaptic plasticity result in modulations of those after-effects. This makes it plausible to assume that the observed after-effects are, in fact, due to synaptic plasticity. Due to the large number of participants/patients that are required for studies on genetic polymorphisms, the evidence is sparse in DBS, which requires the implantation of electrodes in patients, as compared with TMS/tES. 

The three methods of brain stimulation reviewed here operate at different time scales. The duration of rTMS, especially at high stimulation frequencies, is limited to the order of several minutes due to the relatively high amount of energy that is delivered to the brain; tES can be applied for up to about 30 min continuously due to its reduced energy as compared with TMS; DBS is typically applied chronically over years. Nevertheless, it would be interesting for future studies to directly compare the three described methods with each other regarding their effects upon synaptic plasticity.

The three methods also differ with regard to their focality. DBS is the most focal method with stimulation electrodes directly inserted into brain tissue, thereby, directly stimulating neurons in their vicinity. rTMS is a little less focal, since the magnetic field generated by the coil has to penetrate the skull before inducing an electric field inside the brain tissue. This electric field is strongest in superficial brain areas and decreases in intensity in deeper brain areas. tES is the least focal of the three methods. The electric field has to penetrate the skull and reaches all the way from one stimulation electrode to the other. For conventional, two-electrode montages, the maximum of the resulting electric field inside the brain occurs in the area between the two electrodes. For more advanced montages using a smaller area for current injection and a larger area to return the current, the field can be focused in the proximity of the injecting electrode. If the electrodes are placed too close to each other, the electric current is shunted by the scalp and only little current reaches brain tissue. While tES methods allow for stimulating superficial brain regions, targeting deep brain regions is not possible without strong co-stimulation of the overlaying cortex. A relatively new method, transcranial temporal interference stimulation (tTIS), has been developed in an animal model and aims to avoid this disadvantage of tES [244]. For tTIS, two pairs of electrodes are placed on the scalp, each introducing a banana-shaped region of current density inside the brain. Sine waves of slightly different frequencies are fed into the brain via each pair, for example, 1000 Hz and 1010 Hz, both frequencies being outside the frequency range relevant for brain activity, i.e., above 1000 Hz. In these brain areas where the two regions of current flow overlap, the two frequencies interact and a beat frequency can be seen at the difference frequency, i.e., 10 Hz. Simulations of the electric fields during tTIS suggest that this approach can target deep brain regions, whilst substantially reducing co-stimulation of the overlaying brain regions [245]. In addition, simulations of computational neural network models suggest that such beat frequencies are, in principle, capable of engaging neural oscillations [246,247], albeit at much higher stimulation intensities as compared with conventional tES methods. Notably, a first study has recently demonstrated that this method is able to modulate human brain activity [248]. Future studies should evaluate whether plasticity can be induced by this method. 

### Limitations

The literature that we cited revealed partially conflicting results. This is most likely due to small sample sizes of the studies. Sample size becomes particularly an issue when studies attempt to relate neuroimaging results of a few subjects to heterogeneous clinical scores which can vary widely even within the same subject. In that case, we focused on reporting imaging and electrophysiological evidence. In addition, we decided not to review literature on animal research that investigated plasticity directly at the synaptic level. Instead, we focused on human studies applying neurostimulation. This decision limits our conclusions since only animal studies can unequivocally demonstrate synaptic changes. Human studies on neurostimulation can only observe indirect effects of neuroplasticity. In contrast to animal models of brain stimulation, brain stimulation in general, and specifically in DBS, works over a period of years to decades [249,250], which cannot be recreated in an animal study. It has to be noted that after-effects observed after neurostimulation could also result from other mechanisms than neuroplasticity. Potentially, neurostimulation could result in other effects such as up- or downregulating the secretion of neurotransmitters such that altered levels of these neurotransmitters outlast the end of stimulation. Especially in the case of altered behavior, indirect effects of neurostimulation are conceivable. For example, if PD patients experience improved motor function during neurostimulation, it can be assumed that they, in turn, move more after neurostimulation. In that case, the observed after-effects could also be due to increased mobility.

In order to demonstrate more unequivocally that the abovementioned after-effects of brain stimulation are, in fact, due to neuroplasticity, future studies should focus on the involvement of relevant neurotransmitters, receptors, genes, etc. [251]. For example, positron emission tomography (PET) is feasible in parallel to all three brain stimulation methods described in this review (TMS [85,89], DBS [252,253], and tES [254,255]). In the past, however, PET was mainly used to investigate how brain activity changes in response to brain stimulation, i.e., regional cerebral blood flow was assessed [256]. It is, however, also possible to investigate how brain stimulation changes the binding of very specialized ligands to certain neurotransmitters and their receptors [257]. Crucially, a recent study used PET imaging to visualize AMPA receptors in humans [258]. A combination of PET imaging and brain stimulation would further our understanding of the interplay between brain stimulation and neuroplasticity.

## Figures and Tables

**Figure 1 brainsci-12-00929-f001:**
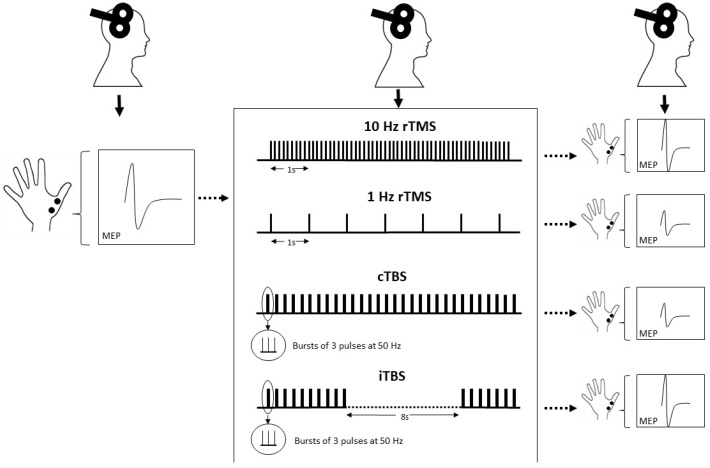
Test amplitudes are elicited by single-pulse TMS before and after application of rTMS protocol and can be used as a measure of cortical excitability. The effect depends, among other factors, on stimulation frequency and pattern. rTMS at high frequencies (e.g., 10 Hz) increases cortical excitability, while low-frequency rTMS (e.g., 1 Hz) decreases cortical excitability. TBS is a patterned form of rTMS and decreases cortical excitability when applied as a continuous train and increases cortical excitability when applied intermittently, i.e., 2 s repeated every 10 s.

**Figure 2 brainsci-12-00929-f002:**
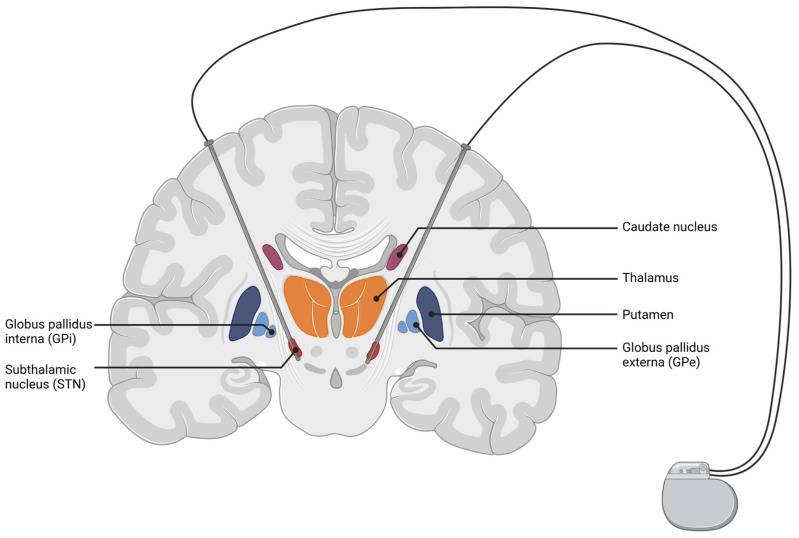
Coronal cut of the brain, highlighting structures of the basal ganglia (BG) with an exemplary depiction of a DBS setup targeting the subthalamic nucleus. Electrical current is delivered from the implanted pulse generator to the targeted brain structure. The figure was created with biorender.com (accessed 1 February 2022).

**Figure 3 brainsci-12-00929-f003:**
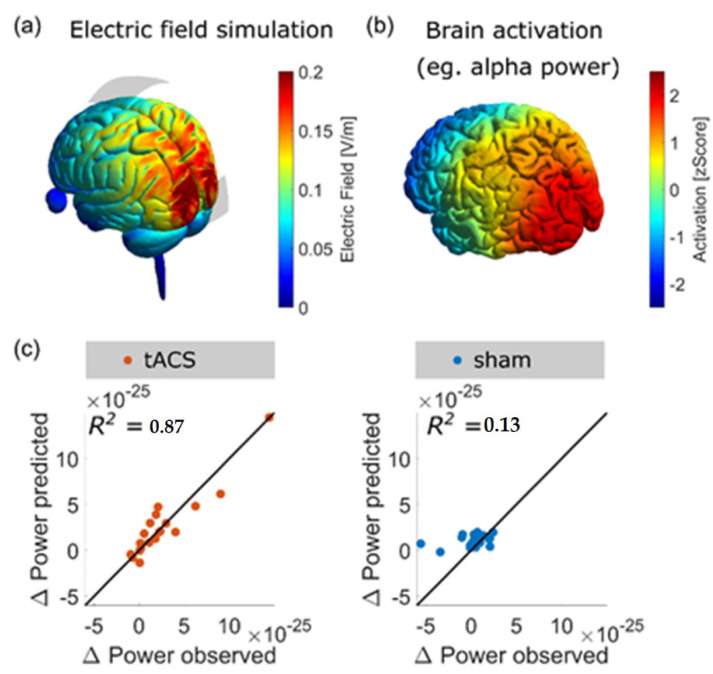
(**a**) Visualization of current density pattern for a montage with stimulation electrodes at EEG locations Cz and Oz; (**b**) source localization of the human alpha generator from a MEG experiment; (**c**) it was recently shown that properties of the electric field (i.e., the similarity of the electric field and the activation map and the strength of the field) can be used to model the expected effect of an alpha-tACS experiment aiming to increase the amplitude of the alpha activity. Only after active (but not sham) stimulation the model predicted changes in alpha amplitude. Adapted from Ref. [185].

**Figure 4 brainsci-12-00929-f004:**
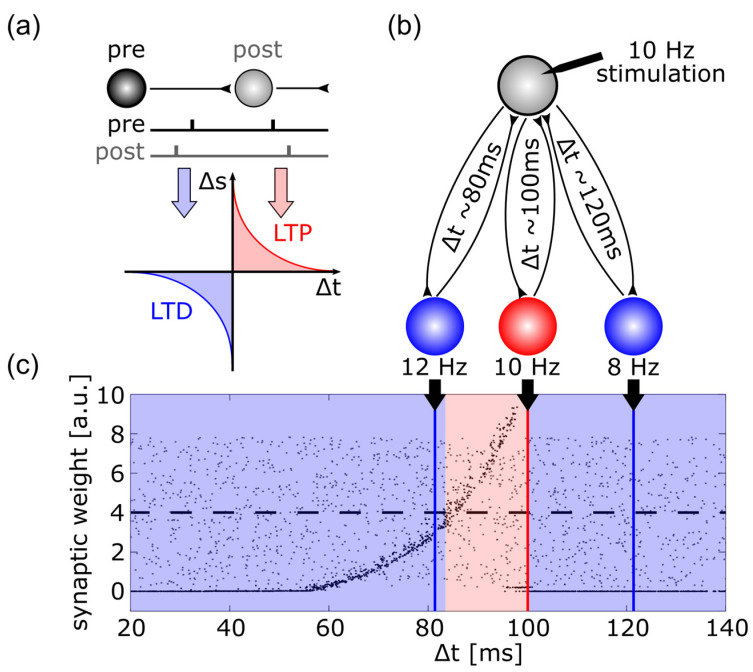
This figure illustrates the mechanism of spike-timing dependent plasticity (STDP) which might explain after-effects of transcranial brain stimulation. (**a**) Synaptic weights are increased if a post-synaptic potential follows a pre-synaptic spike, i.e., long-term potentiation (LTP) occurs, if, however, a post-synaptic potential occurs prior to a pre-synaptic spike, long-term depression (LTD) is the result; (**b**) schematic illustration of a network simulation: A driving neuron (gray) established a recurrent loop with each neuron of a hidden layer, the total synaptic delay (i.e., the sum of both delays of the loop) varied between 20 and 140 ms (only three such loops are shown here), the driving neuron was stimulated with a spike train of 10 Hz; (**c**) synaptic weights of the back projection as a function of the total synaptic delay of the recurrent loops: grey dots display synaptic weights at the start of the simulation ranging from 0 to 8, black dots represent synaptic weights after the end of simulation. External stimulation of the driving neuron at 10 Hz resulted in synaptic weights above the initial average of 4 (dashed line) for recurrent loops with a total delay between 82 and 100 ms, indicating LTP (region shaded in red). For all other delays, synaptic weights were reduced, indicating LTD (region shaded in blue).

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
