# Peer review of "Evidence of Neuroplastic Changes after Transcranial Magnetic, Electric, and Deep Brain Stimulation"

_brainsci, 2022, doi:10.3390/brainsci12070929_

Round 1
Reviewer 1 Report
Dear Authors,
This is a good effort but it suffers from many minor to major issues.
First the reason behind a general review like this is not explained. There are so many reviews on this matter, plasticity, that it is really difficult to understand why a new one is needed. Then the review is citing very old paper, where amazing steps have taken place the last 20 years. This review might has been valid before that.
I have tried to put my comments on the pdf document (attached) but there is so much work to be done that i cannot do it for you. I went up to the middle of TMS. There are many inaccuracies. There are many references missing. It really needs a lot of work.
You need to clarify what plasticity you want to review, from M1 only?You mix everything and it is very difficult to follow and support.
Please rewrite it by having a specific target. State it. You state things that are not true. Please go through the recent litarature. So many reviews, original papers and meta analysis are there. Especially for TMS there is an amazing amount of work that has been done. Please go through and focus so i can direct.

Reviewer 2 Report
This article reviewed evidences on three modalities of brain stimulation techniques; repetitive transcranial magnetic stimulation (TMS), deep brain stimulation (DBS), and transcranial electric stimulation (TES). Authors especially focused on their neuroplastic capacity in human brain. They reviewed stimulation-related neurobehavioral effects of these modalities as well as lasting after-effect measured by neurophysiologic and imaging modalities such as EEG, MEG, EMG, SPECT, and MRI. Possible different neuroplastic effect of these modalities according to the genetic polymorphism was also suggested. Their review is thorough and comprehensive including recent evidences and perspectives of brain stimulation for neuroplasticity. I believe that this article is worthy to give insights to the readers on future researches and clinical applications of various brain stimulation modalities for enhancing succesful neuroplasticity.
Reviewer 3 Report
See attached. Thanks!

Round 2
Reviewer 3 Report
Although I believe that the manuscript has been significantly improved with the edits added, I unfortunately feel that the manuscript still requires significant revisions before consideration of publication.
My main concern at this stage is the organization of the manuscript. The order in which the modalities (TMS, DBS, tES) are introduced in the introduction does not match the order they are described in the manuscript. Please correct this. Additionally, it seems that the review falls somewhere between an overview of evidence in certain sections, and a "deep dive" into the literature in other sections, in a way that may misrepresent the literature. For example, PET studies are described for TMS only - why are PET study findings not described for DBS? Another example - why is the TMS section divided by research modality (pharmacologic studies, behavioral studies, etc.) yet the DBS section is divided by patient group studied, and the tES section is divided by subtypes of tES. This lack of consistency I believe leads to a skewed representation of the literature for each section. I certainly understand the challenge of trying to review all 3 of these modalities in extensive detail in a review article, and appreciate the ambition of the authors attempting to do this, but standardizing the scope across sections would greatly improve the article in my opinion. It appears the authors did may some edits to subsection headings in this edited version which are a good start, but inadequate in addressing this issue.
A few other minor points:
1) The abstract refers to DBS being used to excite or inhibit tissue (line 22) - this is an overly simplistic way of describing its action, in my opinion. Especially in the abstract I would recommend a more nuanced explanation of what it is doing ("modulating" tissue?).
2) Again, I have concerns about several grammatical errors in the text, especially in the edited and added text. Thank you for your efforts to fix grammatical errors in the original draft and please review the edited text to fix these errors.
Author Response
On behalf of my co-authors, I would like to thank the reviewer again for the time and effort they put into reviewing our updated manuscript. We found the feedback is constructive and helpful yet also challenging. Below we elaborate on how we have revised the manuscript and elaborate to the issues raised.
Please find our point by point response below. Our replies to your comments are printed in bold.
Kind regards,
Julius Kricheldorff
Response to Reviewer 3
Response to general comments
My main concern at this stage is the organization of the manuscript. The order in which the modalities (TMS, DBS, tES) are introduced in the introduction does not match the order they are described in the manuscript. Please correct this.
Reply: We agree that this was not an optimal structure and have now reordered the introduction to fit the overall order of the methods which are reviewed.
Additionally, it seems that the review falls somewhere between an overview of evidence in certain sections, and a "deep dive" into the literature in other sections, in a way that may misrepresent the literature. For example, PET studies are described for TMS only - why are PET study findings not described for DBS? Another example - why is the TMS section divided by research modality (pharmacologic studies, behavioral studies, etc.) yet the DBS section is divided by patient group studied, and the tES section is divided by subtypes of tES. This lack of consistency I believe leads to a skewed representation of the literature for each section. ... standardizing the scope across sections would greatly improve the article in my opinion.
Reply: We agree with the reviewer that the individual structure of the different section has been a challenge. However, during conception of this paper and initial research we realized that due to the different intended purposes of these neurostimulation methods, a thorough comparison will be a challenge. The body of literature concerning these three methods is somewhat disparate and therefore no easy to break down. Whereas TMS is a hybrid method that is applied both in the clinic and experimental settings, DBS is an exclusively clinical application and TEs is mostly an experimental application and much fewer and more preliminary clinical applications are available at the moment. As such, evidence for neuroplasticity comes from different application contexts and we did our best to find a structure balanced for both readability, comprehensiveness and comparability. An example, it seems rather unethical to test neuroplastic properties by means of NMDA antagonists in a sample of diseased patients with chronic DBS versus it being legitimate in young healthy participants in a tES or TMS stimulation study. By selecting a clinically-guided structure for DBS and a technically-guided structure for the tES and TMS section, we believe to capture and present the literature better than by using a single one-fits-all structure. As proposed, we restructured the introduction so that it now reflects the individual structure of the sections in the order that they are introduced. Moreover, we have included an introduction-paragraph for the reader highlighting and justifying the individual section structure.
For the method of tES, PET studies have been carried out. However, the studies applied glucose PET to monitor the changes of brain activity rather than to study neuroplasticity. In our revision, we point out that PET is feasible with all methods, refer to several studies and suggest that the field would profit from future PET studies that specifically investigate whether neurotransmitters and/or receptors are modulated by brain stimulation.
We understand and appreciate the issues raised by the reviewer and have revised the manuscript accordingly. Due to the heterogeneous nature of the three methods compared, a review-suggestion which was received with enthusiasm by the editor, we have to compromise between individual section organization and overall inclusion of findings. We hope the new manuscript will reflect the effort we have made to give the reader a good and thorough introduction but keep the manuscript brief and readable and at the same time.
The abstract refers to DBS being used to excite or inhibit tissue (line 22) - this is an overly simplistic way of describing its action, in my opinion. Especially in the abstract I would recommend a more nuanced explanation of what it is doing ("modulating" tissue?).
Reply: We agree that an excitation or inhibition due to DBS is overly simplistic. Now, we used the suggested terminology and refer to a modulation of neuronal activity. Line 22 “Electrodes are placed in the brain in order to modulate neural activity and to correct parameters of pathological oscillation in brain circuits such as their amplitude or frequency.”
Again, I have concerns about several grammatical errors in the text, especially in the edited and added text. Thank you for your efforts to fix grammatical errors in the original draft and please review the edited text to fix these errors.
Reply: We checked the manuscript for remaining concerning grammar and apologize for any remaining errors. For better readability, we did not apply track changes to spelling corrections.
